# A General Computational Framework for COVID-19 Modelling with Applications to Testing Varied Interventions in Education Environments

Joshua W. Moore [1,*], Zechariah Lau [1], Katerina Kaouri [1], Trevor C. Dale [2] and Thomas E. Woolley [1]

1    School of Mathematics, Cardiff University, Cardiff CF24 4AG, UK;
     LauZY@cardiff.ac.uk (Z.L.); KaouriK@cardiff.ac.uk (K.K.); WoolleyT1@cardiff.ac.uk (T.E.W.)
2    School of Biosciences, Cardiff University, Cardiff CF10 3AX, UK; Daletc@cardiff.ac.uk
*    Correspondence: MooreJ16@cardiff.ac.uk

**Abstract:** We construct a spatially-compartmental, individual-based model of the spread of SARS-CoV-2 in indoor spaces. The model can be used to predict the infection rates in a variety of locations when various non-pharmaceutical interventions (NPIs) are introduced. Tasked by the Welsh Government, we apply the model to secondary schools and to Further and Higher Education environments. Specifically, we consider student populations mixing in a classroom and in halls of residence. We focus on assessing the potential efficacy of Lateral Flow Devices (LFDs) when used in broad-based screens for asymptomatic infection or in 'test-to-release' scenarios in which individuals who have been exposed to infection are released from isolation after a negative LFD result. LFDs are also compared to other NPIs; we find that, although LFD testing can be used to mitigate the spread of SARS-CoV-2, it is more effective to invest in personal protective equipment, e.g., masks, and in increasing ventilation quality. In addition, we provide an open-access and user-friendly online applet that simulates the model, complete with user tutorials to encourage the use of the model to aid educational policy decisions as input infection data becomes available.

**Keywords:** education policy; health policy; airborne transmission; stochastic simulation; interventions; agent-based modelling

## 1. Introduction

When the COVID-19 pandemic started sweeping across the globe [1–3], governments tried to stem spread by imposing national lockdowns during which essential sectors, such as education, were shut down, or moved online [4]. Following successive epidemic waves, there have been extensive efforts to get students, across all age ranges, back into schools, colleges and universities [5–8]. Primarily, the arguments for reopening education environments have focused on the benefits of education for the development of skills and knowledge as well as for the maintenance of good mental health [9–12].

There has been a vigorous debate about the role of schools and universities in disease transmission and, as such, a range of non-pharmaceutical interventions (NPIs) have been proposed and implemented in the education sector [13–15]. In the UK, the focus has been on using Lateral Flow Devices (LFDs) as a means of detecting infected individuals to better manage infection [16–18]. It has been suggested that a negative result from an LFD might be used to discontinue the self-isolation of people who are a close contact of an infected individual [19–21], with trepidation caused by the fact that LFDs have a high false-negative rate [22]. We, thus, investigated how beneficial LFDs can be in comparison to other NPIs, such as wearing masks or improving ventilation, particularly as recent literature has reported that despite its success the vaccination programme has led to a reduction in compliance to contact and isolation policies with cases again rising [23]. Moreover, vaccinated individuals can still contract and transmit the infection,

especially if the individual has underlying health conditions and/or is taking medications that impair the immune system [24,25]. The Centers for Disease Control and Prevention state that such an individual should "should continue to take all precautions recommended for unvaccinated people, including wearing a well-fitted mask, until advised otherwise by their healthcare provider" [26]. Hence, there is still an urgent need for continuing to study and compare NPIs in order to identify their optimal use, with emphasis on educational environments.

There have been recent efforts to quantify the use of LFDs in a 'test-to-release' strategy by means of a national experiment of 204 UK secondary schools over a six week period [27]. The study compared the 'test-to-test' strategy against the existing alternative of close-contact isolation, measuring the number of infections confirmed by a positive PCR test within each of the schools. The authors conclude that the use of LFDs in a 'test-to-release' isolation policy did not significantly increase the number of infections within the selected cohorts and therefore suggested its viability to reduce absences from education. However, the study was conducted when the infection was at its lowest prevalence and least transmissive in England in 2021 [28,29]; therefore, it is not clear if the results of the study apply to a highly infectious environment, given, especially, the estimated high false-negative rate of self-administered LFDs [22]. In addition, the number of infections in both the control and intervention groups where scaled against the local community cases, yet due to the high asymptomatic rate of cases for people aged 12–18 [30], the effects of greater prevalence would not be observed within the schools but in the households of the students. However, students with a household member that was isolating from COVID-19 symptoms were excluded from the study. Although, the 'test-to-release' study of Young et al. [27] provided positive preliminary data on the effects of LFDs in classrooms, further analysis is required to understand their use in a variety of infectious environments before promoting 'test-to-release' as an effective strategy to prevent transmission.

Mathematical modelling offers a wide variety of techniques to predict the impact of interventions on infection spread without risking anyone's health [31]. Notably, many different mathematical fields are able to provide predictive results, for example, deterministic and stochastic differential equations [32], machine learning [33], Monte Carlo simulations [34] and queuing theory [35]. For the small number of individuals we are considering a deterministic approach would not be justified [36]. Further, machine learning tools, or continuous time data techniques [37], are not currently viable due to lack of sufficient data. Infection modelling and analysis has been conducted on the impact of school reopening on local $R$ numbers [38]. Here, a novel, detailed methodology for tracking SARS-CoV-2 spread through two, linked spatial compartments that is flexible enough to compare a variety of NPIs has been developed and presented.

We develop an individual-based model of infection spread that splits the susceptible population into two compartments: 'local' and 'non-local' groups. The algorithm can be used to predict the spread of an infection in many situations where individuals naturally form groups, or 'cliques'. We focus on modelling an educational setting; in particular, we seek to predict how testing, isolation and other NPIs influence the spread of infection in secondary schools, Further Education (FE) settings (e.g., colleges) and Higher Education (HE) settings, (e.g., classrooms or halls of residence). Note that the local versus non-local distinction can also be motivated by the presence of short-range airborne transmission due to large droplets and longer-range airborne transmission transmission through aerosols [39].

Our results utilise current UK data [40]. We also provide the reader access to an online applet in addition to a maintained repository of open source MATLAB code (see Section 2.4) that not only reproduces our results, but can also be easily adapted by the reader to include data which are more accurate and/or specific to their location and needs. The online applet can be found at https://bit.ly/CV19_INTER_IBM, accessed on 20 November 2021.

Our results have been presented to the Technical Advisory Group (TAG) of the Welsh Government informing policy on the COVID-19 response in Wales and to the Wales Further and Higher Education COVID-19 Task Group. Our methodology is currently

being used to inform policy in relation to the future of students returning to educational environments. Furthermore, our findings have been shared with the Environmental Science TAG Subgroup for use in advising how to open up more general social spaces, such as places of worship. The results have been communicated across the governments of England, Scotland and Northern Ireland, and have informed the wider development of policy planning for the pandemic.

Section 2 presents the assumptions and processes that underlie the model that controls the individual agents (aggregations of pupils, students, etc.) that we are interested in and simulating. In Section 2.1, we summarise the spatial airborne transmission model we use to estimate the infection risk in a classroom and give some details about ventilation. We clarify how testing and isolation of individuals (agents) is enacted in Sections 2.2 and 2.3. The algorithm is then applied in Section 3 for two settings: (i) a classroom in a secondary school/FE environment, Section 3.1, and (ii) a halls of residence at a Higher Education establishment, Section 3.2. Critically, for each of these two broad applications we consider the effects of applying LFD testing. Finally, in Section 4, we condense our findings down to simple observations regarding the applications of LFD testing versus other possible NPIs.

## 2. A Computational Framework for Estimating Transmission with NPIs

We have constructed an individual-based, stochastic algorithm that predicts infection spread in educational settings. This can be applied to, but not limited to, the COVID-19 pandemic. The model is comprised of independent modules that can be turned on, or off, as appropriate, so a large variety of realistic situations can be explored. For example, we can easily vary the days and frequency in which testing is conducted and the size of student groups. We provide an overview of the algorithm in Figure 1; a detailed flowchart of the algorithm is provided in Appendix A.

We consider a population of $N$ susceptible individuals split into uniform groups of size $N_g$. The parameter $N_g$ is the number of local contacts an individual has and represents the number of individuals that would be instructed to isolate should a member of a group is identified as infectious. If $N_g = 1$ only an infected individual is isolated upon (i) becoming symptomatic, or (ii) receiving a positive LFD test. Whereas, if $N_g = N$, the entire population is isolated upon the identification of an infected individual. Finally, if $N_g$ is any other divisor of $N$ then only a subgroup of the population is isolated upon the identification of an infected individual.

Here, we study various types of educational settings and we ascribe different definitions to $N_g$. In a secondary/FE classroom setting, $N_g$ will represent a 'table group'. These are individuals who are socially distanced according to current regulations, but are sat on the same table. In the HE case, we consider $N_g$ to be the size of a 'kitchen group' (dormitory). These are individuals in shared accommodation in a halls of residence that share the same facilities.

The effectiveness of isolation is modulated by a 'compliance parameter', $C$. In the secondary school and FE classroom settings we assume that the school or college is keeping records of positive tests and has a final say on who is allowed to enter the classroom. Thus, we assume that $C = 100\%$ and, any student that is symptomatic, or receives a positive test result, is isolated completely for a set amount of time in which they cannot pass on further infections to the simulated population.

However, for the FE and HE cases, information supplied by the Welsh Government's Head of Policy HE COVID-19 (B. Cradock, 4 February 2021, pers. comm.) suggests that students in the HE environment are only 81% likely to comply with an isolation order. Hence, in such cases, after receiving a positive test result, or becoming symptomatic, a HE student is only isolated with a probability specified by the parameter $C$.

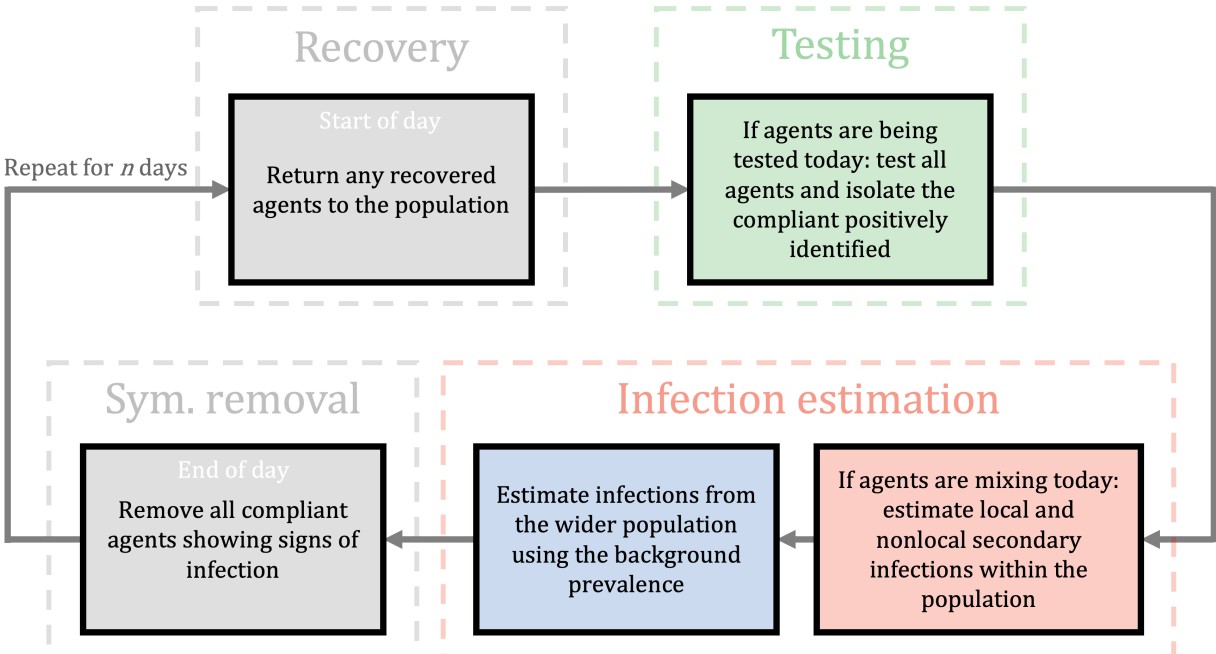

**Figure 1.** A flowchart of the algorithm outlining the computational framework. It illustrates the general iterative process controlling the dynamics of a population over a typical day. Grey boxes represent actions that occur every day whereas the coloured boxes can be independently switched on/off on any prescribed day. A single cycle accounts for a single day. We start each day with recovery (see recovery box) which returns any isolating agents that have completed their isolation period. Then, if testing is active, the individuals are tested using LFDs and those agents with a positive test are removed from the population for $t_r$ days (see testing box). If agents are allowed to mix, then we estimate secondary infections (see infection estimation boxes) using the effective $R$, as obtained using appropriate scalings (see Section 2.1.1). Next, if external infection sources are being considered, we estimate the number of infections to the susceptible population using the background prevalence of the disease (see infection estimation boxes). Finally, we remove any compliant symptomatic individuals (see sym. removal box) that display the infection at the end of every day. For a detailed flowchart of the infection algorithm, see Appendix A.

As discussed above, our decision-making framework accounts for both symptomatic and asymptomatic individuals. The percentage of symptomatic individuals is defined as $P_s$ and at the point of infection each individual is either given a symptomatic, or an asymptomatic indication based on this percentage. Asymptomatic individuals are able to infect other susceptible individuals but will not isolate unless tested, whereas symptomatic individuals are assumed to follow public health guidance and follow self-isolation guidance at a rate determined by the compliance parameter $C$. Symptomatic individuals who choose not to be tested are treated as asymptomatic. A non-exhaustive list of real-world infection events and model interpretation is provided in Appendix A, Table A1.

The timescale of the model is one day. For every day we specify whether individuals are to be tested and whether they are able to mix. Specifically, we say individuals in the population are mixing if they are able to interact with each other, for example, if students are in the classroom, they are considered to be mixing. The testing and mixing scenarios are independent and, thus, we control whether neither, one, or both scenarios occur. For example, because we are modelling a school, or a college, we currently specify that secondary and FE students do not mix on a weekend and, equally, no testing happens on a weekend. On the other hand, in the halls of residence related to the HE simulations mixing and testing can occur every day.

As show in the flowchart in Figure 1, "Testing" box, during any day that testing is applied all individuals that are not isolating are tested. The testing phase occurs before the mixing phase. Thus, any individual triggering a positive result is isolated before they can infect others. Critically, we currently do not include infection transmission once individuals

are isolating as we suspend any isolating agent from the simulation for $t_d$ days. Under this assumption, in a secondary school, if the whole class is isolating no further infections appear. This assumption is reasonable in the secondary and FE cases as the students are physically isolated from one another. However, in the HE case, further infections may occur within isolated flats. Thus, when we compare the isolating individuals across those that are infected and those that are not, we only quote this ratio to be correct at the point of isolation, because, as mentioned, further infections within an isolating subgroup are not tracked but are likely to occur—this could be a future work direction; see Section 4.2.

As the model can contend with subgroups within a population, we have developed an infection model with a two-compartment spatial complexity where the agents are either in the 'local' or in the 'non-local' compartment. The mechanism of dynamically altering individual interactions based on pair-wise positioning has been widely applied and analysed in other biomedical scenarios [41–43] but it has yet to be employed extensively for studying SARS-CoV-2 transmission between individuals in small populations.

This addition of spatial complexity, enables the split of the basic reproduction number, $R$, into two contributions:

1. A local number, $R_l$, that measures the expected number of secondary infections within a subgroup that has an infected person; and
2. A non-local number, $R_n$, that measures the expected number of secondary infections that occur in the non-local compartment further away;

$R = R_l + R_n$ where the basic reproduction number, $R$, can be estimated from current data. Specifically, we define the ratio of local to non-local infections as $c_l$:$c_n$, and, consequently, $R_l = c_l R / (c_l + c_n)$ and $R_n = c_n R / (c_l + c_n)$.

The value of the ratio $c_l$:$c_n$ depends on how we assume that the virus spreads. For example, in a secondary school classroom environment, if there is a lot of movement, talking, and inter-group mixing then we might expect there to be no difference between the number of infections in groups to the number of infections outside a group; in such a case $c_l = c_n$ and $R_l = R_n$. However, if an effective social distancing policy is implemented, we would expect the number of non-local infections to be lower than the number of local infections, i.e., $c_n < c_l$ and consequently $R_l < R_n$.

For the secondary school classroom scenario, we determine $R_l$ and $R_n$ using the airborne infection risk estimated by modifying the model in [44]. In [44], the concentration of airborne particles in a classroom is determined by solving an advection–diffusion–reaction equation (see Section 2.1). Subsequently, the *infection risk* is determined by modifying the well-known Wells–Riley formula [45,46]. Applying the same methodology here, we evaluate the local *infection risk*, $c_l$, at two metres directly downwind from an infected individual in the centre of the room with low level activity (20% talking) and for three ventilation settings. This represents the worst-case scenario as it maximises the amount of contagion that a susceptible person would receive if there was an infected individual in their subgroup while a 2 m social distancing was strictly observed. Further, we evaluate the non-local *infection risk*, $c_n$, at a distance of four metres, in the direction orthogonal to the airflow from an infected individual. This represents the average amount of contagion that a susceptible person would receive if there was an infected individual in a different subgroup. The airborne transmission simulations are run for five hours as we assume this to be a reasonable amount of time spent in the classroom in a day to determine $c_l$:$c_n$.

In the halls of residence (HE) scenario, we choose to define the ratio $c_l$:$c_n$ differently, using the compliance of the individuals to a social distancing policy. Specifically, we consider two cases. The first is the 'poor social distancing' where residents from all flats mix and local and non-local infection numbers are equal ($c_l = c_n$). The second case considers 'enhanced social distancing', where on average, five out of six flat residents are only socialising with members of their same flat, whilst one in every six people still mix with other flats even when strict social distancing measures have been advised. Thus, in the enhanced social distancing case, the ratio of local to non-local infections is 5:1. This value was estimated to agree approximately with the isolation compliance probability of

$C = 81\%$. However, we did not simply choose a ratio of 4:1, which would match this compliance, because the ratio of 5:1 is more interpretable when the flat occupancies are of size 6, or 12, which are characteristic HE halls of residence capacities in the UK. Thus, on average, there is one or two poorly compliant individuals in the flats, respectively.

We now consider the blue and red boxes in Figure 1, which depict the 'infection estimation' steps of the algorithm. The blue box controls external infections, namely infections that occur outside the classroom, or halls of residence. In the classroom scenario, we assume that these happen at the end of the day due to socialising after school, which is why the red box feeds into the blue box. These infections occur at the local prevalence rate, $I$ [47], and can be turned on or off, depending how well we believe our system is isolated from the outside world. Equally, isolating the system allows us to study how the system influences itself under any intervention. We consider all populations to be closed communities in each of the scenarios considered in this study. Namely, we assume that members of the population (students) mostly interact with each other and therefore the background prevalence does not play a role, i.e., the blue box of Figure 1 is inactive (B. Cradock, 4 February 2021, pers. comm.). Although, scenarios in which individuals regularly interact with members from outside of their community can be analysed using the provided online applet (see Section 2.5). In such scenarios, we expect increased transmission particularly as the model assumes that the infection events from wider communities are independent, that is, external factors such as household immunity are neglected.

The red box in Figure 1 controls how secondary infections are generated due to mixing between individuals. During any day where the students can mix then any individual that is:

- infected;
- infectious;
- not isolated;

can infect other people. Once a person becomes infected, three 'delay clocks' are attached to each of them. The three delays [48], $t_i$, $t_d$ and $t_r$, represent:

- The time between becoming infected to becoming infectious; we take $t_i = 3$ days,
- The time between becoming infected to becoming detectable by a LFD test; we take $t_d = 5$ days, and
- The time between becoming infected to recovery; we take $t_r = 10$ days.

Any recovered person is assumed to be non-infectious and immune to further infections. Currently, these delays are fixed, but these could easily be made variable. We assume the recovery and isolation timescales to be independent, so, once an individual is discovered to be infected, they are isolated for 10 further days, no matter where they are within the infection timeline. For example, if an individual is found on the 9th day of their infection, they have to isolate for another 10 days, even though they would recover the next day.

We assume that the tests being used are LFDs, which are assigned to have a false-positive probability, $P_{fp}$ and a false-negative probability, $P_{fn}$. In a recent systematic review of LFD efficacy in SARS-CoV-2 detection, the authors summarised the results of twenty-four studies involving over 26,000 RT-PCR confirmed cases comparing the sensitivity and specificity of LFDs from global manufacturers [49]. The authors conclude that LFD specificity ranged from 92.4% to 100%. In particular, the manufacturer Bioeasy was highlighted as the only LFD supplier with an average specificity <97% [49]. These results are in agreement with the study conducted by the UK COVID-19 Lateral Flow Oversight Team which was designed to analyse the efficacy of LFDs suppliers to the UK [50]. Specifically, the study considered 64 suppliers including Innova, Deepblue, Roche and SuresScreen, analysing the specificity of the LFDs using a total of 6954 RT-PCR confirmed negative samples. The authors conclude a high specificity of >97% over the accumulated data set. Moreover, another independent UK study conducted by Pickering et al. [51] found an average specificity >98% when comparing six UK suppliers (Innova, E25 Bio, SureScreen visual (V), spring, Encode

and SureScreen fluorenscent (F)) each examining 100 negative samples [51]. In particular, SureScreen visual (V) and Encode achieved 0% false-positive rate whereas the comparative E25 Bio assays contained 14% false-positive rate. Although, the authors remark this as an anomaly as all E25 Bio LFD detected positives were weak (projected Ct values of <25). Accordingly, throughout all simulations, we fix the false-positive probability fixed to $P_{fp}$ = 0.003, in agreement with the aforementioned UK LFD studies [50,51], although, this value can be adjusted using the provided online applet.

Furthermore, the systematic review of LFD efficacy conducted by Mistry et al. [49] found that LFD sensitivities ranged from 37.7% to 99.2%, highlighting that LFD false-negative rates are not only dependent on the manufacturer but also on the operator, and thus we consider the probability of LFD false-negative result to be variable.

### 2.1. Interventions

#### 2.1.1. Scenario: Classrooms

In order to quantify the effects of ventilation, and masks and other environmental factors on transmission of SARS-CoV-2 in classrooms, we employ an airborne transmission model that consists of an advection–diffusion–reaction (ADR) equation and a formula for estimating the infection risk (probability of infection), as modified from the model in [44]. The model extends the well-known Wells–Riley model for the transmission of airborne infectious diseases [45,46] by taking into account that viral concentration changes also with space. With this model we can quickly estimate the concentration of infectious particles and the associated infection risk in an indoor space. The ADR equation is of the form

$$\frac{\partial A}{\partial t} + \nabla \cdot (\vec{v}A) - \nabla \cdot (K \nabla A) = S_{\text{inf}} - S_{\text{vent}} - S_{\text{deact}} - S_{\text{set}}, \tag{1}$$

where $A$ is the concentration of airborne viral particles (particles/$m^3$) and $t$ is the time (s). Equation (1) models viral particles advected by airflow, diffused due to turbulence, emitted by infected people ($S_{\text{inf}}$) and removed by three processes; on the right hand side, $S_{\text{inf}}$ represents the source of infectious particles (i.e., the infectious people) and $S_{\text{vent}}, S_{\text{deact}}, S_{\text{set}}$ are sink terms that represent, respectively, the removal of particles by ventilation, biological deactivation, and gravitational settling. One asymptomatic or presymptomatic infectious person who breathes or talks is considered and who may wear a mask. In this works we always assume an average mask efficiency of 50% which we consider valid as we cannot ensure all individuals have correctly fitted masks and may occasionally remove their masks etc. Under appropriate assumptions (see Appendix C), Equation (1) has a semi-analytic solution that allows for swift simulations and quantification of the effect of ventilation and masks on transmission.

The infection risk, $P_A$, is estimated using the dose-response formula

$$P_A(x, y, z, t) = 1 - \exp\left(-k \int_0^t \rho A(x, y, z, \tau) \mathrm{d}\tau\right), \tag{2}$$

at any location in the classroom, where $k$ is an constant that depends on the infectiousness of the virus, and $\rho$ is the average breathing rate. The infection risk (2) calculates how likely is for a susceptible person to be infected. Further details can be found in Appendix C and in [44].

The epidemiological $R$ value can be modified to take into account various NPIs, for example, masks [52], ventilation [53] and portable air purifiers [54]. Ventilation is the provision of fresh outside air into the room, diluting and displacing the concentration of indoor contaminants, including infectious particles [55]. Ventilation may be introduced naturally—through openings such as doors and windows—or mechanically, for example through an air conditioning unit. The amount of ventilation recommended for an indoor space is dependent on its function, occupancy and size [55]. In our airborne transmission model we assume mechanical ventilation and based on realistic scenarios and

recommended ventilation settings by ASHRAE (standards and guidlines are issued by the American Society of Heating, Refrigeration and Air-Conditioning Engineers) [55], we consider three ventilation settings: 'very poor ventilation' (ACH = 0.12 $h^{-1}$, i.e., closed windows and no A/C or fans active) [56], 'poor ventilation' (ACH = 0.72 $h^{-1}$) [56] and the ASHRAE recommended minimum ventilation setting 'good ventilation', (ACH = 3.00 $h^{-1}$, i.e., A/C or fans active) [55,56]. Using the model (1) and (2), we estimate the ratio of viral particle concentrations and infection risk between 'local' and 'non-local' individuals using a prototypical classroom environment, for each of these three ventilation settings (see Appendix C). We assume that masks halve the emission rate of viral particles [57]. We also assume 'low activity', which we take to mean that individuals speak only 20% of the time (the remaining time they stay silent, only breathing). By comparing the *local infection risk* during 'low activity', we assume that the reduction in the *R* value is equal to the assumed efficiency of the masks, in this case 50% (see Appendix C, Tables A6 and A7). In addition, we assume that the *R* number that is parameterised using the national *R* number (estimated using real data) [58] corresponds to 'very poor' ventilation (ACH = 0.12 $h^{-1}$), as we assume that this is the case in many classrooms in the UK, especially in the winter when no windows can be opened and A/C is not available or active. By comparing the local infection risk at 'low activity' in the three ventilation settings, we can scale the *R* number accordingly. For instance, a good ventilation setting yields a 3.5-fold decrease in *R*. The combined influence of masks and ventilation is assumed to be multiplicative and, thus, *R* is reduced seven times. For all the scalings, please see Appendix C, Table A7. A summary of combinations used in classroom simulations can be found in Table 1.

Improving ventilation also decreases the ratio of the local infection risk to the non-local infection risk, since better ventilation facilitates better mixing of the air in the room and thus lower viral concentrations [59]. In contrast, masks have a localisation effect as they prevent viral particles from propagating significant distances [60]. The combination of ventilation and masks in classroom environments has been studied in detail using the model (1) and (2) and the results can be found in Appendix C. Throughout our simulations, we assume that the individuals exhibit low activity (intermittent talking, 20% of the time) and we apply the best-case and worst-case ventilation scenarios (see Table 1). Furthermore, we assume that the classroom environment has very poor ventilation and no mask usage when obtaining the *R* value from data. In Appendix C, Table A7, we summarise the values of the ratio of the local to the non-local infection rate.

**Table 1.** A summary of infection parameters generated by Equations (1) and (2) for individuals doing 'low activity' (only 20% talking overall) in a classroom environment, for the three ventilation settings considered For details on parameter estimations, see Appendix C.

| Environment Scenario | Effective $R$ | Local to Non-Local Infection Ratio ($c_l$:$c_n$) |
|---|---|---|
| Very poor ventilation + no masks | $R$ | 3.98:1 |
| Very poor ventilation + masks | $R/2$ | 5.20:1 |
| Good ventilation + no masks | $2R/7$ | 1.47:1 |
| Good ventilation + masks | $R/7$ | 1.51:1 |

2.1.2. Scenario: Halls of Residence

In the halls of residence scenarios ventilation cannot be improved without significant expense—cheaper solutions like portable air purifiers could be considered [61]. Further, masks are usually not worn whilst individuals are in their own flats. Thus, we assume that the enacted interventions are reducing the number of non-local contacts an individual has, following guidance from the university and the government. Specifically, for the HE halls of residence scenarios, we do not change *R* directly as interventions are introduced; instead we change the ratio of the number of local to non-local infections. In the so-called 'poor social distancing' scenario, residents from all flats mix and, thus, the local and non-local infection numbers are equal, that is $c_l$:$c_n$ is 1:1, whereas in the 'enhanced social distancing' scenario, individuals are encouraged to isolate as much as possible, and they, more likely,

socialise only within their flat group ('kitchen group'). This reduces the number of non-local infections, compared to the local infections; we thus, take the ratio $c_l$:$c_n$ to be 5:1/5, for reasons stated above, in Section 2.

2.1.3. Vaccinations

At the time of conducting this study, vaccination of people aged 12–16 had not been implemented in the UK [62]. In addition, the present study focuses on NPIs and therefore we do not consider vaccinations as a method of transmission mitigation in both the classroom and the HE halls of residence simulations of Section 3. However, immunity via vaccination or recent infection has been implemented into the online simulator of the model (https://bit.ly/CV19_INTER_IBM, accessed on 20 November 2021)—see details on this in Appendix B. By including the option of vaccination in the applet, the model may be applied to explore infection in educational settings as youth vaccination is rolled out. Moreover, vaccinated individuals can still contract and infect other individuals, particularly if an individual has underlying health conditions [25]. Hence, these vaccination effects on transmission may be accounted for by an appropriate scaling of the infection reproduction number $R$, and therefore generalises the applications of the present model to non-educational settings in different age groups.

*2.2. Testing Regimes*

Alongside varying the false-negative probability of the LFD, we consider a variety of different testing and isolation scenarios. The base case against which we compare all other cases is the no testing and no isolation of contacts case. Namely, the disease is simply allowed to spread through a class, or halls of residence without impedance.

Noting that in the cases modelled, only non-isolating individuals use LFDs and we can alter the testing regime for the population from weekly to daily, and any specified schedule. Further, we can alter the event that causes testing to occur. Either testing occurs on a fixed regime, or alternatively, no testing is applied until a first symptomatic person is found, that is, there exists an agent showing symptoms that day (a trigger event). Once a symptomatic individual is found, testing is applied through a weekly, or daily regime.

Other "reactive" testing regimes can be included (e.g., you always test after a symptomatic is found, rather than just moving to a fixed regime), but the "no testing" to "daily testing" range provides the extreme limits in which all other possible testing scenarios must fall. In Section 3, we will see that these extreme testing scenarios provide limits over a small range of results and, thus, all other proposed testing regimes must fit within these limits. Hence, understanding only the limiting cases provides us with enough knowledge to understand all cases.

As a specific example, in the secondary school case, we are particularly interested in the case of "test-to-release". Namely, it was previously the case that if an infected individual was found, then their close contacts would be isolated too—in this case, this would be their table group; see Figure 2c. However, current guidance [5–7] is that a negative LFD result could release individuals in the table group that test negatively. Under our simulation, this would conform to the individual isolation case. Namely, only the individual is forced to isolate, whilst everyone else is tested before they are then allowed to mix; see Figure 2b.

In the HE setting, we compare the efficacy of the various testing strategies at two time points during a term. In the first scenario, called the 'start of term' scenario, the first simulated week occurs prior to students returning to the halls of residence. In this initial week, students do not mix, but any infected student will incubate their infection. The next three weeks are simulated under the assumption that the students are back in their halls and able to mix daily. The second scenario, called the 'middle of term' scenario, considers a period post student arrival, where the individuals are mixing each day (including weekends) for four weeks.

### 2.3. Isolation Regimes

As mentioned in Section 2.2, we initially consider the case in which no isolation occurs. This situation can then be compared against:

- Isolating individuals due to the individual being symptomatic, or receiving a positive test result;
- Isolating subgroups due to at least one individual in the subgroup being symptomatic, or receiving a positive test result;
- Isolating an entire population due to at least one individual being symptomatic, or receiving a positive test result.

Note that the positive test results leading to an isolation can be either true or false positives.

Throughout the simulation, we keep track of the number of people who are isolating. Further, the scatter points illustrated in Section 3 are all pie charts that represent the ratio of infected people who are isolating (white part of the pie chart) versus the number of people who are isolating, but healthy (colour part of the pie chart).

### 2.4. Robustness of LFD Testing Strategies: Parameter Estimation

To elucidate the robustness of the LFD testing strategies in various infectious environments, we simulate the mathematical model across a range of parameter values. We consider the best-case scenario to have the following parameter values: a low background prevalence, $I$; a low $R$ number, and a low probability of false negatives, $P_{fn}$. The situation becomes worse when any of these parameters is increased. For each parameter, we estimate two extreme values (see Table 2) and simulate over all the nine combinations arising. In addition to these global parameters, we provide a summary of context specific parameter values and parameter definitions in the Tables A2 and A3 of Appendix A. Note that the $R$ number, as referenced in Tables 2 and A3, is the local background $R$ number, parameterised using the national $R$ number estimates for our best-case and worst-case scenarios [58]. The actual numbers that apply to education scenarios as the ones considered here are anticipated to lie between these extremes and may depend on whether transmission between children, adolescents and young adults differ substantially from aggregate national $R$ numbers [63]. Clearly, local strategies such as increasing ventilation or mandating mask use may have already contributed to the lower estimate of the National $R$ number. Nonetheless, we anticipate that the effect of the modelled NPIs is included between the two extreme values of the $R$ number in Table 2.

**Table 2.** Table of parameter values used in all education environments simulations.

| Parameter | Definition | Best-Case Parameter Values | Worst-Case Parameter Values |
|---|---|---|---|
| Probability of false negatives, $P_{fn}$ | Likelihood that a SARS-CoV-2 test does not identify an infectious individual. | 0.2 | 0.5 |
| R number, $R$ | The average number of secondary cases we would expect an infected student to generate prior to locality affects. | 0.8 | 1.7 |
| Background prevalence, $I$ (%) | Additional probability that more than one person is infected. | 0.5 | 2.0 |

### 2.5. Open-Source Code and User-Friendly Applet

Our aim is to illustrate the use of our computational framework in selected educational settings of interest but the algorithm can be easily used to predict infection numbers across a wide range variety of other scenarios. Further, we provide one interpretation of how interventions influence the outcome. The algorithm could be used to provide predictions when new interventions are introduced in educational settings (e.g., portable air purifiers),

so long as the user has formed a clear idea on how the interventions influence the input parameters (e.g., the *R* number).

Moreover, we explore a wide range of reasonable parameter values illustrating how diverse our results can be. As parameter estimates are becoming better over time, leveraging the more data becoming available, the simulations can be easily run with updated parameter values.

An online COVID-19 Intervention Simulator has been developed, dedicated to secondary school and FE environments has been developed based on the framework presented here and it has been made publicly available—see (https://bit.ly/CV19_INTER_IBM, accessed on 24 November 2021). The applet allows the user to both reproduce the results of the study here and also to design new infection scenarios with current infection data without the required coding ability or other software. Appendix B contains information on usage and features. In addition, all source code for the model and applet is stored and maintained at https://github.com/joshwillmoore1/COVID-19_Intervention_IBM, accessed on 20 November 2021. The results shown in this study were run and tested on MATLAB 2020b and the applet was developed using R v4.0.3. We hope that providing this user-friendly simulator of our model and making the source code openly available will enable others to expedite their response to the pandemic through exploring which of the available to them intervention options are the best for their location.

## 3. Results

The algorithm is able to generate a large number of outputs. Specifically, it generates the infection, isolation and symptomatic status of every individual every day. Because the algorithm is stochastic, the code is run 1000 times for each scenario and this provides us with an understanding of the sensitivity of our estimates. The code outputs many daily statistics but below we choose to visualise only the average total number of infections versus the average number of days in isolation per individual.

Note that the speed of the simulations is mainly determined by the size of $N$, since we fix the number of repetitions. For the FE classes, which are of size $N = 10$, running 1000 simulations takes approximately 9.9 s. In the scenario for the halls of residence, where $N = 204$, the results take approximately 40.3 s (all simulations were run on a 2.6 GHz 6-Core Intel Core i7 with 16 GB 2667 MHz DDR4 2019 MacBook Pro.). As we sweep over the 5-dimensional parameter space required to analyse the best-case and worst-case scenarios, simulation times scale linearly, achieving a maximum simulation time of approximately four hours for the HE settings.

As mentioned, in Section 2.3, we choose to use pie charts to track numbers of infected or healthy individuals isolating and place them in the space of 'average total number of infections' versus 'days of isolation per individual'. Each pie chart represents a different intervention. The intervention is encoded in the colour and transparency of the pie chart. The coloured part of each pie chart represents the proportion of 'correct' isolations, i.e., isolations of infected people, whereas the white section represents the proportion of incorrect isolations, i.e., isolations of healthy people. Thus, when seeking to optimise the choice of NPIs for a particular scenario of interest, we may independently assess overall infection rates and strategies that minimise the isolation of uninfected individuals using both distance from the origin and proportions of the pie charts.

Overall, we assume that policy makers are primarily interested in minimising the total number of infections and the number of days spent in isolation. Thus, we look to strategies that achieve this. When these two goals are in conflict, it would be assumed that minimising the total number of infections would have priority.

### 3.1. Secondary School Environments

In this case, we set the class size to be $N = 30$ and consider $N_g = 1, 5$ and $30$. Thus, apart from the case where no isolation occurs, we will have: (a) only an infected individual

isolating and their table group, or (b) the entire class isolating, respectively (see Table A1 for our interpretation of real-world events in the modelling framework).

As a base case for investigating the efficacy of LFD testing we, first, run the model without testing included. Thus, the parameter $P_{fn}$ is irrelevant. Thus, we focus on varying the level of symptomatic prevalence, $P_s$ between 20% and 50%. Having a large population of asymptomatic individuals, potentially causes problems when we run a reactive testing strategy, rather than a fixed testing strategy (see Section 3.1.2) [64,65]. When testing is considered, we fix $P_s = 20\%$, which provides the worst-case scenario of 80% of the infected population not presenting symptoms.

Equally, from simulating all combinations of the best-case and worst-case parameters as discussed in Table 2 we note that reducing $R$ reduces the average total number of infections and the average number of days isolation better than reducing the false-negative probability, $P_{fn}$. Noting that this observation remains true over all simulations, for clarity, we only present the results for the best-case and worst-case scenarios.

The simulations are run over 28 days and we assume that the simulation starts on a Sunday; see Figure A1 for an explicit description of simulation initialisation. Defining the starting day is important as we assume that testing and mixing can only occur during weekdays, not weekends.

In a secondary school, the Head Teacher has the authority to remove any student displaying symptoms from the classroom. In addition, as the school oversees testing and instructs students to isolate accordingly, we set the percentage of student compliance to 100%. Although, varying compliance and background infection rates, within realistic limits, (i.e., $60 \leq C \leq 100$ and $0.5 \leq I \leq 2$) does not significantly influence the results. Specifically, reducing the local prevalence rate $I$ to 0.5% from 2% has an insignificant effect on the number of infections within the classroom due to the population size being small ($N = 30$). We would have required at least a prevalence rate of $I = 3.3\%$ to infect at least one student per simulation. Further, although changing the compliance has an obvious influence on the quantitative results, the qualitative results remain the same from the perspective of intervention efficacy. Hence, in the following figures, we choose a background infection prevalence of $I = 2\%$ and a compliance of $C = 100\%$.

### 3.1.1. Without Testing

Figure 2 presents the simulations under the assumption of no LFD testing. Namely, isolations can only occur when a student becomes symptomatic (3 days post infection) and/or when those who have been exposed to possible infection are asked to isolate.

Figure 2a presents the worst-case scenario where individuals are exposed to a presymptomatic infectious individual but no isolation is implemented, for $R = 0.8$ and $R = 1.7$ and with $P_s = 20\%$ or 50%. Thus, if there are no other interventions, eventually everyone becomes infected within a four-week period. Note that, since there are no isolations, no healthy people are isolated and hence the pie charts are fully coloured. In Figure 2b, only the single symptomatic infected individual isolates, whereas, in Figure 2c,d, the school has been notified and either the table group, or the entire class, respectively, has been asked to isolate.

We observe from Figure 2a–d that when the level of intervention is increased, fewer people become infected, as one might expect. Moreover, although combining mask use and improving ventilation is the best policy, if we were to choose a single intervention to apply, increased ventilation is more effective than mask use.

We also note from Figure 2b–d that increasing $R$ increases the average number of infected individuals, as expected (in Figure 2a, all individuals are infected after 28 days even for the lowest value of $R = 0.8$). Moreover, an increase in the percentage of individuals who display symptoms ($P_s$) from 20% to 50% reduces the total number of symptomatic individuals who self-isolate (compare Figure 2a,b), where the characteristic values of $P_s$ represents the extreme bounds of percentage symptomatic of the 12–16 age group [30]. As an indirect consequence, a reduced number of infections feeds through into a lower

average number of days in isolation per individual (Figure 2b). A reduction in *R* similarly leads to a reduction in both infection numbers and the days of isolation. Note that we are unable to physically influence the probability of an infected becoming symptomatic, $P_s$, as this is an intrinsic property of the infection. Therefore, in subsequent figures, we only focus on varying *R* as this is the parameter that NPIs can influence.

**Figure 2.** Average total number of infected students and average number of days in isolation per individual when no testing is employed, over different isolation strategies and types of interventions, in a secondary school classroom. Each colour represents a different intervention—see legend at the top of the figure for details and Section 2.1 for a description of how each intervention influences the parameter values. (**a**) No isolation. (**b**) Single isolation. (**c**) Table group isolation. (**d**) Class isolation.

Another result that is consistent across all our simulations and across the secondary, FE and HE scenarios is that isolating larger subgroups of the population is one of the surest ways of reducing the average total number of infections—see Figure 2b,d. The contribution of contact isolation becomes greater when good ventilation and mask use is not in place. However, this strategy increases both the number of days in isolation and the number of healthy individuals isolating. Specifically, as the isolation group size is increased from just the individual, to the table group and, finally, to the entire class the pie charts in Figure 2b–d move left, up and their non-coloured part increases.

### 3.1.2. With Testing

Figure 3 represents the same intervention strategies as those shown in Figure 2 but we also consider multiple testing scenarios. As mentioned, testing frequency is denoted by pie chart *transparency*. Namely, as the testing frequency is increased, the pie charts become more transparent. Note that the no-testing results from Figure 2 are also depicted as the opaque pie charts, to allow for easy comparison between no testing and various testing strategies.

The simulations appearing in the left column of Figure 3 all assume that testing only occurs after a first symptomatic individual is detected ("reactive" testing), whereas the right column is for a testing strategy that is fixed at the beginning of the simulation. For both reactive and fixed testing strategies, we consider once-a-week testing and daily testing. We see that the difference caused by these two strategies are minimal; they may, however, differ in the cost of implementation which is not studied here. The reason is that, even at a symptomatic (to asymptomatic) prevalence of 20%, a symptomatic individual is generated extremely quickly through secondary infections, which causes a reactive testing strategy to kick in quite soon after a fixed strategy (random testing for asymptomatic students).

The overall trends mentioned in Figure 2 remain the same when comparing the different levels of testing, namely: reducing $R$ is the best way of reducing the number of infections; ventilation is better than masks as a single intervention where their combination is more effective than each one alone; and that isolating more people greatly reduces the overall number of infections but increases the number of days in isolation (absences) and the number of healthy people being isolated.

Notably, we see that the addition of weekly, or daily tests does reduce the number of overall infections, but more tests lead to more absences as more cases are found earlier in the infection cycle (the pie charts move left and up as they become more transparent). Critically, although testing does help, we see that reducing $R$ is much more important, particularly due to the time-lag between being infectious and being detectable by LFD ($t_d > t_i$) and so testing will never be able to fully remove infections from the population, under our assumptions. As a result, wearing a mask and having no testing is as good as daily testing without a mask in the majority of the test scenarios we have considered (the opaque red pie chart is always lower and left of the most transparent black pie chart).

When considering 'test-to-release' strategies, we note that infection rates were lower when students are asked to isolate if they have been in contact with infected individuals. Namely, in Figure 3e, where $R = 1.7$ and $P_{fn} = 50\%$, the black, red, blue and green rates (solid colour; no testing) at 14, 11, 7 and 3, respectively. By contrast, the equivalent numbers for students who are tested and released back to the School Population in a 'test-to-release' strategy (in Figure 3a, $R = 1.7$ and $P_{fn} = 50\%$, the black, red, blue and green rates (transparent colour; daily testing) are 28, 26, 20 and 5. This would suggest that, if the primary policy goal was to minimise infections, 'test-to-release' would increase infections by comparison with either the table-isolation or entire-class isolation strategies.

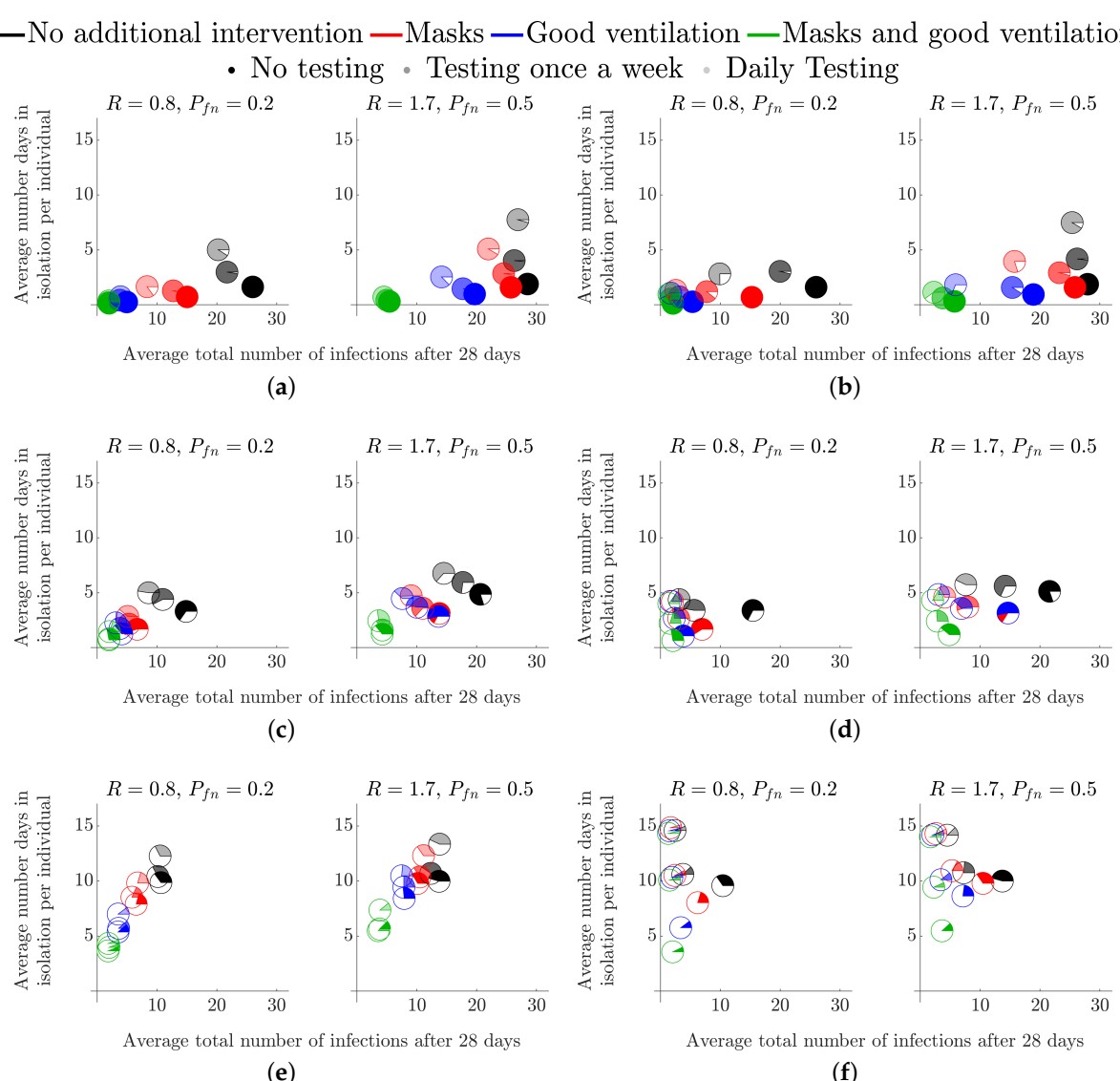

**Figure 3.** Average total number of infected students and average number of days in isolation per individual when testing is employed, over a variety of isolation strategies and interventions in a secondary school environment. Each colour represents a different intervention—see legend at the top of the figure for details and see Section 2.1 for a description of how each intervention influences the parameter values. (**a**) Single isolation, testing from first symptomatic. (**b**) Single isolation, testing fixed from the beginning of the simulation. (**c**) Table group isolation, testing from first symptomatic. (**d**) Table group, testing fixed from the beginning of the simulation isolation. (**e**) Class isolation, testing from first symptomatic. (**f**) Class isolation, testing fixed from the beginning of the simulation.

### 3.2. Higher Education Environments

The cohort size in this case is the population of a typical halls of residence, $N = 204$. These individuals are separated into flats, or 'kitchen groups' of size $N_g = 6$, or $N_g = 12$. We fix the symptomatic prevalence to be $P_s = 40\%$ and the compliance to be $C = 81\%$ to account for the 18–24 age group and social settings [30].

Here, we mainly focus on elucidating the influence of the flat size on the results and timing of the test. Specifically, see Section 2.2, where we discuss the start of term and middle of term testing scenarios.

#### 3.2.1. Start of Term

The start-of-term simulations start a week prior to student mixing. Hence, all initially infected individuals are potentially detectable by the first administered LFD test in both strategies. Therefore, the lag time between being exposed to the virus, being infectious and

being detectable has no effect on the initial transmission within the student population. In this scenario we consider the effect of a test prior to returning to the halls of residence. Namely, in Figure 4, the blue pie charts represent the case in which students are tested two days prior to arrival (Friday of week 1), followed by weekly tests, every Monday, starting on the day of arrival (Monday of week 2). The red pie charts represent the case where there is no test before returning. However, weekly tests are administered, every Monday, starting on the day of arrival (Monday of week 2), as in the blue pie chart case.

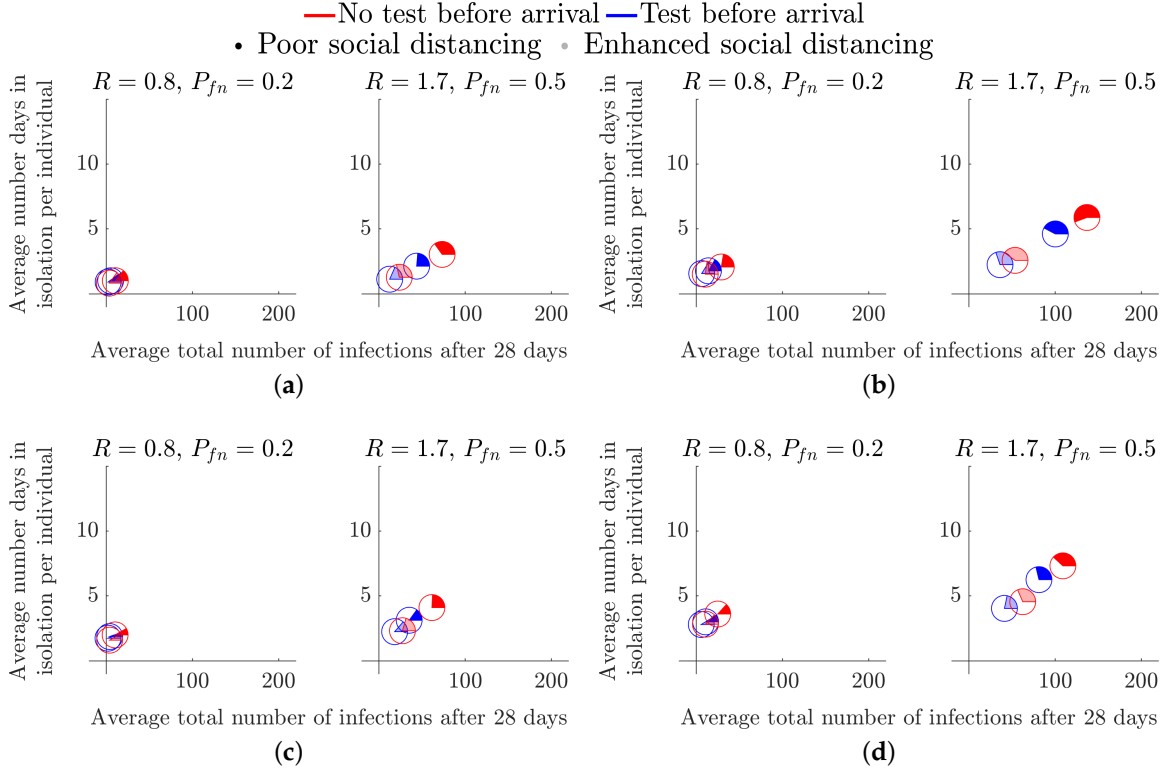

**Figure 4.** Considering the effects of a pre-term LFD test across flats of different occupancy. (**a**) Size 6 flat, $I = 0.5$%. (**b**) Size 6 flat, $I = 2$%. (**c**) Size 12 flat, $I = 0.5$%. (**d**) Size 12 flat, $I = 2$%.

Figure 4 demonstrates that the addition of a test prior to arriving only has a significant benefit in high prevalence and high *R* number environments. The disparity in the total number of infections is particularly prominent in the cases of poor social distancing restrictions as the opaque pie charts are further apart (see Figure 4b,d).

By comparing the results of Figure 4 across flat size, we observe that larger flat sizes result in a small decrease in average total number of infections, but a small increase in the average length of isolation period. Thus, although, a 12-person flat has an increased risk of someone bringing in an infection to the flat, once a positive test result occurs we remove more non-infected people from circulation. However, because we are only tracking infections to the point of isolation, we must be cautious regarding this interpretation. Specifically, once a flat has been isolated, unless all remaining healthy individuals practice extremely good hygiene, it is highly likely that most in the flat will succumb to the infection, meaning that the small reduction in total number of infections that is apparent in the results would not exist and in fact the larger flats would lead to a gain in average total of infections. Critically, this comes down to the responsibility of the individuals of an infected flat. Once a positive test has been received the flat should be clearly informed of their options and best practices that will keep the individuals safe.

In all cases, we see that the coloured part of the pie chart is in the minority. Thus, at the point of isolation, we are isolating more healthy people than infected people. In particular, in larger flat sizes, over 70% of all students isolating in every scenario are healthy in the

12-person flat simulations (see Figure 4c,d). Though it is not substantially less in the smaller 6-person flat, it is worth noting that increasing the flat size increases the number of healthy people isolating.

### 3.2.2. Middle of Term

We next consider the situation where students have returned and are continuously mixing for a 28 day period. In every infection scenario we simulated, we show that increasing the testing frequency greatly reduces the total number of infections (the green and black pie charts are the closest and furthest markers from the origin, respectively, in all subplots of Figure 5). Further, in all subplots of Figure 5, we observe the benefit of including spatial compartments to the model by the restriction of social interactions from the whole population (opaque markers) to 'kitchen groups' (transparent markers), particularly in the cases of $R = 1.7$, as the spatial component of the model considers the allocation of infections between the infected group and the rest of the student population (see Section 2). Critically, when the $R$ number is large, we have a distinct disparity between the opaque and transparent markers, thus, enhanced social distancing measures reduce both the number of infections and the number of days in isolation.

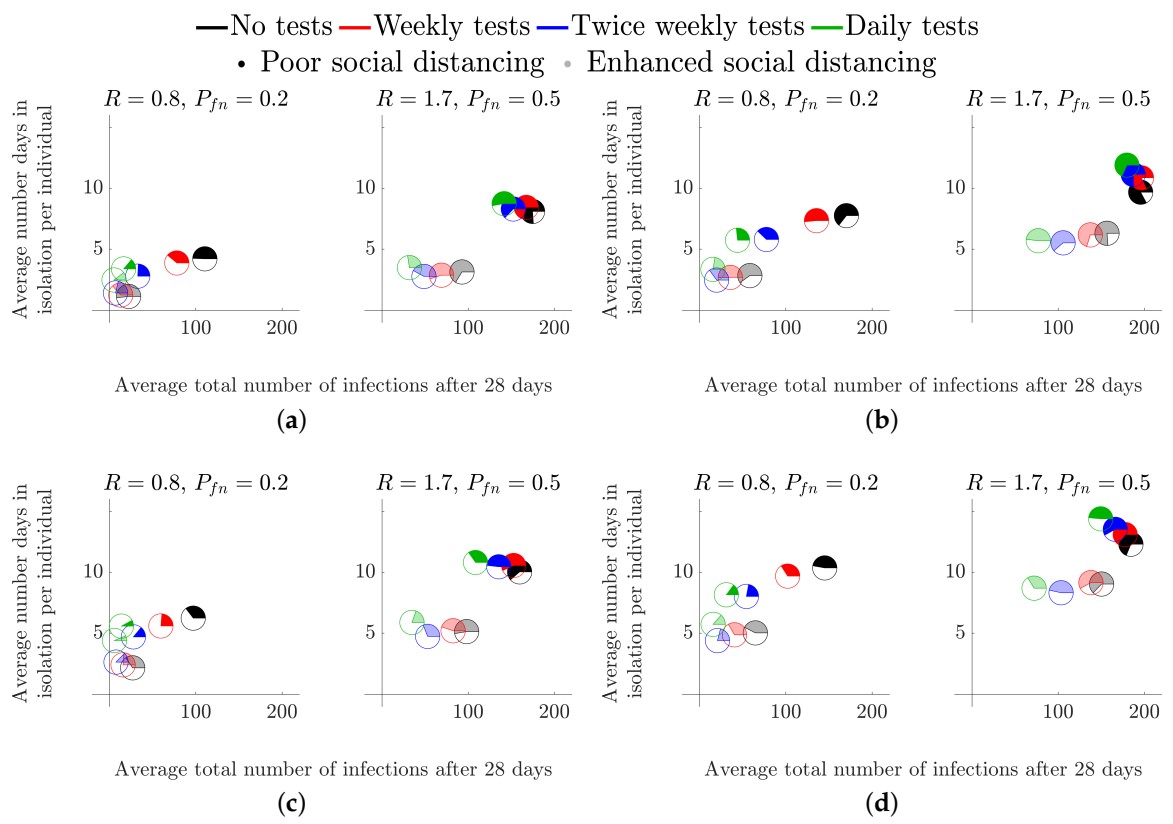

**Figure 5.** Considering the effects of a mid-term LFD test across flats of different occupancy. (**a**) Size 6 flat, $I = 0.5\%$. (**b**) Size 6 flat, $I = 2\%$. (**c**) Size 12 flat, $I = 0.5\%$. (**d**) Size 12 flat, $I = 2\%$.

Critically, as in the secondary school case of Section 3.1, although testing is able to reduce the average total number of infected individuals it is much better to reduce the $R$ value through encouraging enhanced social distancing measures. Specifically, in all cases, the transparent black pie chart is always closer to the origin than the green opaque pie chart.

Finally, our simulations suggest that increasing the flat size improves the efficacy of the enhanced social distancing measures, as all transparent pie charts are closer to the origin in the 12-person flats than their associated pie chart in the 6-person flat simulations. This observation is explained by the fact that increasing the size of a flat increases the number

of healthy students that have to isolate when at least one infected student is found. Again, as mentioned in the previous section, we should critically examine this claim because unless the isolated individuals practice good hygiene the infection will spread through the isolating flat, thus generating more cases than expected. Future iterations of the model will contain this additional ability, see Section 4.2.

## 4. Discussion

Here, we have focused on several interventions across diverse educational contexts, focusing on secondary schools, FE and HE settings. Throughout all of our results, we have seen that the most simple method of reducing infection numbers is to isolate larger groups of individuals. At the extreme, isolating the entire population ensures that infections cannot be transmitted between individuals. Of course, such extreme solutions are unlikely to be desirable because many healthy individuals would be isolated, missing educational and social aspects of everyday life [66]. Further, the ability to apply such a lockdown is impossible as "isolation fatigue" sets in and compliance with harsh rules will likely diminish [67,68].

Thus, how, when and who should be isolated are all important questions that should be carefully considered. Critically, from Figures 3 and 5 isolating individual flats (in halls of residence), or entire classes or table groups (in classroom settings) drastically reduces the average total number of infections. However, this does not dictate total isolation, but encourages effective social distancing to limit infection spread.

We have observed that testing can help reduce the total number of infections as LFDs enable us to find asymptomatic individuals, who would be able to spread the disease, even if they themselves do not become ill. However, due to the time delay between becoming infectious and being detectable by a LFD [48], testing is never going to be able to significantly mitigate infection spread, no matter how often it occurs. This would even be true in the case of PCR tests, which are more accurate (with fewer false negatives) and can produce results earlier in the incubation cycle. However, the time delay in obtaining results from RT-qPCR assays means that there may still be approximately one day between becoming infectious and becoming detectable. The only benefit would be a reduction in the proportion of false negatives ($P_{fn}$), which, as we have seen, is not an important parameter to focus on. Thus, although testing will partially mitigate transmission, we must turn to alternative interventions if we want to have a significant reduction of the spread of the disease.

We also looked at more proactive NPIs that have a financial cost, namely using personal protective equipment (e.g., masks); or investing in improving ventilation units. From the results of Section 3 we concluded that the latter interventions are always going to have positive influence on mitigating the spread, independently of the testing frequency. Moreover, including multiple interventions, such as combining masks and improved ventilation, offers larger gains than each intervention alone. Using airborne transmission modelling [44], we were able to show that although masks of 50% efficiency are a cheap and simple way of reducing disease spread. Improving ventilation from 'very poor' to 'good ventilation' (ACH = 3 h$^{-1}$) would be more effective than mask use. However, it may be the case where more efficient masks perform as well as improved ventilation. We show that the classroom 'test-to-release' isolation strategy may only be viable where there is sufficient ventilation (Figure 3). In addition, even under worst-case parameter values, good ventilation means that we only need to isolate table groups to ensure the greatest reduction in total cases. Hence, we suggest that the most optimal method to mitigate disease spread and educational absences is to target the effective *R* value within educational settings via masks and improved ventilation, in conjunction with infrequent LFD testing for asymptomatic surveillance. However, the economical and social viability of this approach still requires further analysis.

### 4.1. Impact

Our work has already aided the Welsh Government in the development of FE and HE workplace and residential policy. Our model was used to assess the impact of testing and interventions for students returning to colleges, or their halls of residence from their permanent home addresses. Our analysis was considered by the Welsh Government's Technical Advisory Group (TAG), Further Education and Higher Education Task Group dealing with COVID-19. Further, the work was also presented to TAG-E, the Environmental Science Subgroup of TAG, for use in advising how to open up more general social spaces, such as places of worship. Our work has also been communicated to the English Government, Scottish Government and Northern Ireland Executive and is currently being used by Bethan Cradock, (Head of Policy HE COVID-19) and Marian Jebb (Head of post-16 Quality and Data Management for the Welsh Government) to develop policy for safely returning students back to their places of study.

### 4.2. Future Work

Due to the continued existence of the pandemic, there is still plenty be done. Much of the work could focus on making the simulation more accurate. Namely, rather than fixing parameters, we could use a Bayesian approach to sample from realistic parameter distributions, which can be generated from data [69–72]. Equally, current research is focused on generating estimates of real-time simulations of airborne particle spread. This could be encompassed into our simulation, but we would need to increase the time resolution from days to hours.

The quickest gain for making the simulation more realistic, particularly in the HE case, would be to nest the algorithm within itself. Nesting the algorithm within itself would give us access to multiple spatial compartments. Namely, one level of the algorithm could be running a halls of residence on the scale of grouping everyone into flats, then a second level of the algorithm could be running on the scale of individuals within flats. This would allow the removal of the current restriction of tracking infections up until the point of group isolation.

The development of a nested algorithm would also allow us to simulate infection propagation over multiple classes within the same school. As a result, we would be able to elucidate the impact of the teachers moving from class to class, i.e., the existence of "super spreaders" from internal and external class interactions. Namely, in a next iteration, we could track not only when someone is infected, but who infected them. We could then extract the number of secondary infections linked to each infected person and observe whether there are specific individuals that infect others at a rate significantly larger than others. From this point, we could reverse engineer the situations in which the super spreaders find themselves, to see if there are any commonalities which could be perturbed leading to a reduction of their highly infectious nature. Furthermore, the construction a transmission network between all agents would allow for further analysis on the existence of a "super spreader" within the population by determining connectivity bottlenecks from its spectral properties [73].

### 4.3. Summary

We asked whether it was suitable for Lateral Flow Devices (LFD) to be used as a means of getting students back to school, or university. Our simulated results show that repeated testing does help reduce the average number of total infections, as asymptomatic individuals can be found and isolated, resulting in the reduction of infectious individuals. However, we have also found that it is not worth investing in better tests that reduce the false-positive probability of the LFD, which has been its major criticism. Instead, time, effort and money are better spent investing in personal protective equipment (e.g., masks) and increasing the quality of ventilation in enclosed environments.

**Author Contributions:** Conceptualization, T.C.D., T.E.W. and J.W.M.; methodology, T.E.W. and J.W.M.; software, T.E.W. and J.W.M.; validation, Z.L., K.K., T.E.W. and J.W.M.; formal analysis, T.E.W. and J.W.M.; investigation, T.E.W. and J.W.M.; writing—original draft preparation, T.E.W. and J.W.M.; writing—review and editing, T.C.D., Z.L., K.K, T.E.W. and J.W.M.; funding acquisition, T.E.W. and J.W.M. All authors have read and agreed to the published version of the manuscript.

**Funding:** This research was funded by Just One Giant Lab (JOGL) grant number 520772. The APC was funded by Cardiff University's Institutional Open Access Fund.

**Acknowledgments:** J.W.M. is supported by Knowledge Economy Skills Scholarships (KESS2), a pan-Wales higher-level skills initiative led by Bangor University on behalf of the Higher Education sector in Wales. It is part-funded by the Welsh Government's European Social Fund (ESF).

**Conflicts of Interest:** The authors declare no conflict of interest.

## Appendix A. Individual Based Model Interpretation and Implementation

As we developed our computational framework to be used with no coding/technical background, we supply a list of example real-world infection events and how these events are interpreted within our model (see Table A1). We intend for the model to be analysed, adapted and extended to suit the local requirements, thus Table A1 should allow a user to prescribe the inputs required in their particular case.

**Table A1.** A table representing real-world infection events as interpreted by the model framework presented in this study.

| Real-World Event | Infection Classification | Infectious to the Population at Post Event | Maximum Number of Days Infectious Prior to Event (Days) | Maximum Number of Days Infectious Post Event (Days) | Isolation Status Post Event | Susceptible Status Post Event |
|---|---|---|---|---|---|---|
| Presymptomatic student attending school | Asymptomatic | ✓ | 0 | $t_d - t_i$ | ✗ | ✗ |
| Non-compliant symptomatic student attending school | Asymptomatic | ✓ | $t_d - t_i$ | $t_r - t_d$ | ✗ | ✗ |
| Symptomatic student with a true-positive test result compliant with isolation policy | Symptomatic | ✗ | $t_r - t_i$ | 0 | ✓ | ✗ |
| Symptomatic student with a false-negative test result compliant with isolation policy | Asymptomatic | ✓ | $t_r - t_i$ | 0 | ✓ | ✗ |
| Asymptomatic student with a true-positive test result compliant with isolation policy | Asymptomatic | ✗ | $t_r - t_i$ | 0 | ✓ | ✗ |
| Asymptomatic student with a false-negative test result | Asymptomatic | ✓ | $t_r - t_i$ | $t_r - t_i$ | ✗ | ✗ |
| Non-infected student with a false-positive test result from random testing | Non-infected | ✗ | 0 | 0 | ✓ | ✓ |
| Non-infected student with a true-negative test result from random testing | Non-infected | ✗ | 0 | 0 | ✗ | ✓ |

**Table A1.** *Cont.*

| Real-World Event | Infection Classification | Infectious to the Population at Post Event | Maximum Number of Days Infectious Prior to Event (Days) | Maximum Number of Days Infectious Post Event (Days) | Isolation Status Post Event | Susceptible Status Post Event |
|---|---|---|---|---|---|---|
| Asymptomatic student in close contact with a positively identified infected student | Asymptomatic | ✗ | $t_r - t_i$ | 0 | ✓ | ✗ |
| Non-infected student in close contact with a positively identified infected student | Non-infected | ✗ | 0 | 0 | ✓ | ✓ |

Figure A1 provides a flowchart depicting the full algorithm for infection propagation throughout a discrete population, colour-coded in agreement with Figure 1 to highlight optional sub-routines of the algorithm. In addition. we provide Table A2, which presents a summary of parameter values used in each scenario in Section 3. Finally, Table A3 provides the reader with definitions of every input parameter to the model.

**Table A2.** A summary of parameter values used for secondary school and HE simulations. See Section 2.2 for information on various testing regimes.

| | Number of Agents | Testing | Agent Mixing | Wider pop. Infections | Isolation Group Sizes | Probability of Symptomatic | Probability of Compliance |
|---|---|---|---|---|---|---|---|
| **Secondary school (no testing)** | 30 | ✗ | Weekdays | ✗ | 1, 5 & 30 | 0.2 & 0.5 | 1.0 |
| **Secondary school (with testing)** | 30 | ✓ | Weekdays | ✗ | 1, 5 & 30 | 0.2 & 0.5 | 1.0 |
| **Halls of residence (start of term)** | 204 | ✓ | 1 week isolated followed by 3 weeks continuous mixing | ✗ | 6 & 12 | 0.4 | 0.8 |
| **Halls of residence (middle of term)** | 204 | ✓ | Everyday | ✗ | 6 & 12 | 0.4 | 0.8 |

**Table A3.** A table of model parameters with their associated definitions.

| Parameter | Definition |
| --- | --- |
| Probability of false-positive, $P_{fp}$ | The likelihood that a LFD test identifies a non-infected agent with a positive result. |
| Probability of false-negative, $P_{fn}$ | The likelihood that a LFD test does not identify an infected agent with a positive result. |
| R number, $R$ | The average number of secondary cases expected from an infected agent to generate prior to locality affects. |
| Local R number, $R_l$ | The proportion of R that accounts for the number of secondary infections within a subgroup of the population. |
| non-local R number, $R_n$ | The proportion of R that accounts for the number of secondary infections to those outside the infectives subgroup. |
| Background prevalence, $I(\%)$ | The rate at which the wider population introduces an infection to an agent. |
| Probability of compliance, C | The likelihood that an agent is compliant with testing and isolation policies. |
| Probability of symptomatic, $P_s$ | The likelihood that an infected agent will become symptomatic. |
| Total population size, $N$ | The total number of agents in the simulation. |
| Subgroup size, $N_g$ | The number of agents that make up a subgroup within the total population. |
| Infectious time, $t_f$ | The time between becoming infected to becoming infectious. |
| Detectable time, $t_d$ | The time between becoming infected to becoming detectable by a LFD. |
| Recovery time, $t_r$ | The time between becoming infected to recovery from infection. |

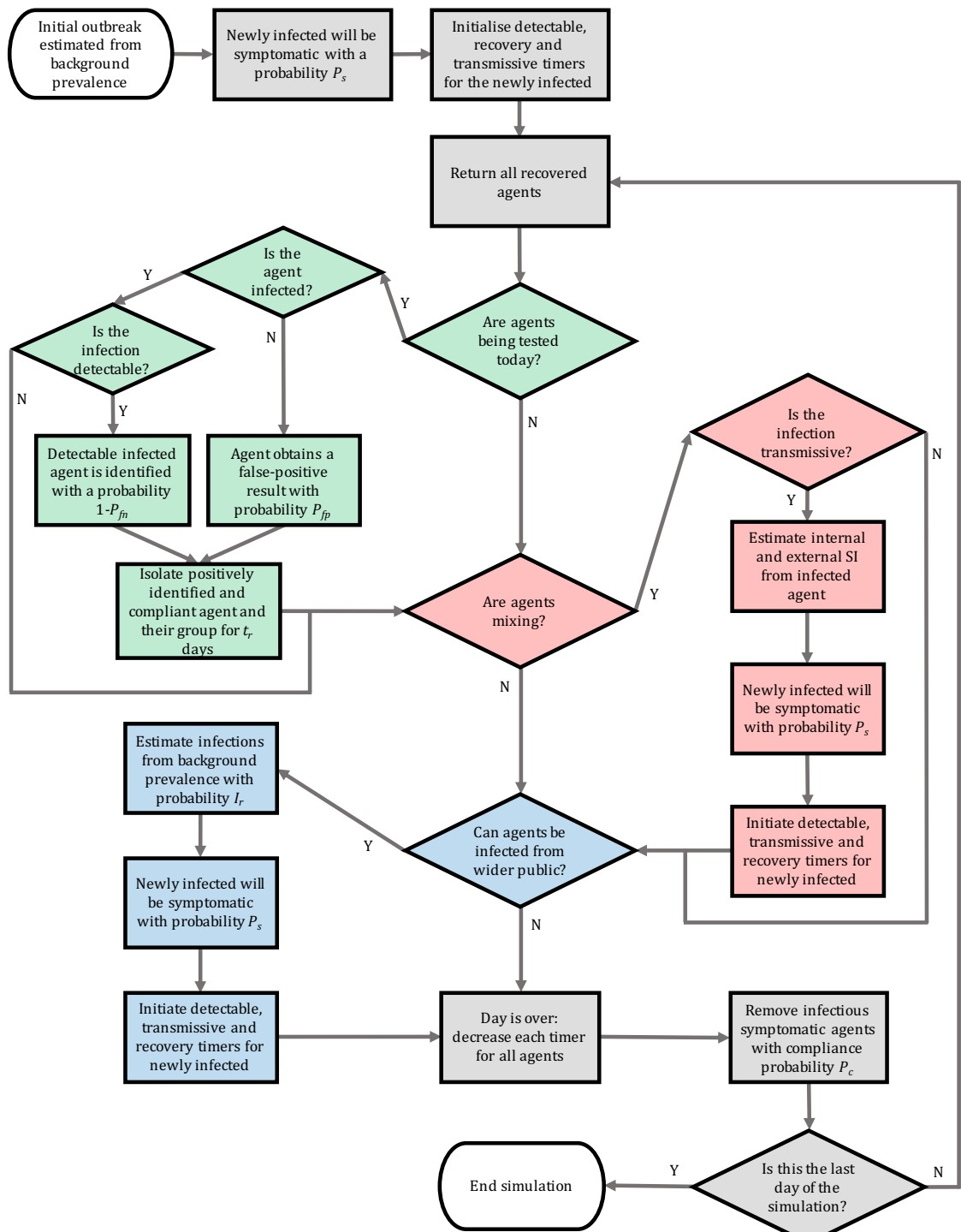

**Figure A1.** Flowchart illustrating the algorithm developed for the individual based model of infection spread. The various subroutines are highlighted using colours: grey is base infection routines, green is testing, red is secondary infection estimation and blue represents infections from outside the population. SI corresponds to secondary infections.

## Appendix B. Open-Access Online COVID-19 Intervention Simulator

An online simulator was developed to encourage the use of the stochastic agent-based model for policy-makers (https://bit.ly/CV19_INTER_IBM, accessed on 20 November 2021). Namely, the applet requires no previous coding experience to operate and allows the user to reproduce the results presented in this study in addition to testing new scenarios with current infection data. The online applet contains all features of the original

model which are outlined in Section 2 with some additional features to aid the decision process of policy surrounding NPIs as the circumstances evolve throughout the pandemic. The additional features of the applet are as follows:

1. Immunity of individuals in the population (vaccinations/recent infections);
2. Optional automatic Welsh infection data retrieval (included to aid the Welsh TAG);
3. Export simulation input data and output data into a downloadable Excel document for further analysis.

The inclusion of these features allow the users to have up to date model predictions as new data is presented, and therefore reduce the time between data collection and policy action.

The applet was developed in collaboration with the Welsh TAG education policy sector to improve usability among policy-makers. In addition, tutorial videos are embedded into the applet to ensure appropriate usage of the software. Figure A2 demonstrates an application of the applet testing the use of masks in a large population.

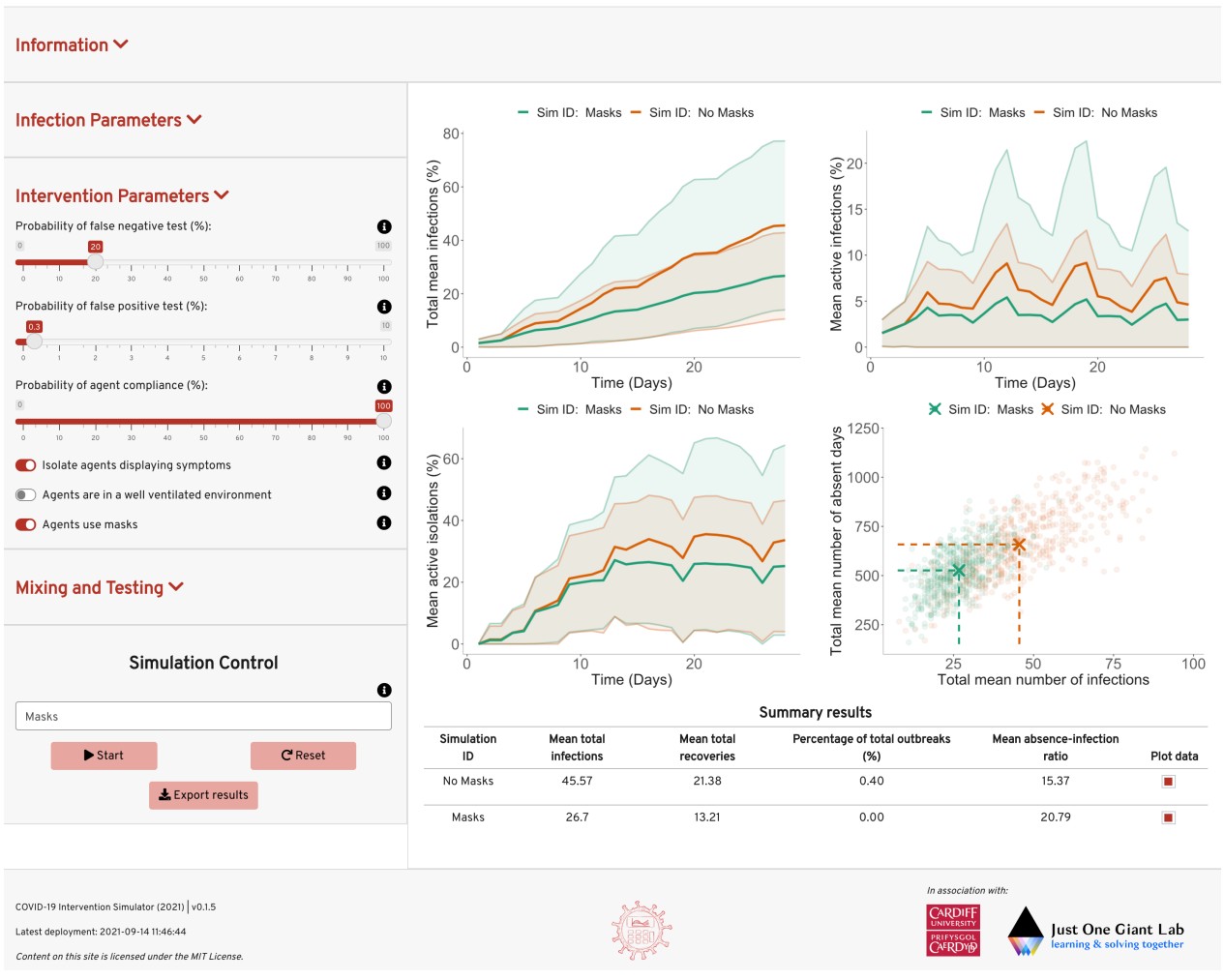

**Figure A2.** An application of the COVID-19 Intervention Simulator to compare the effect of masks on infection transmission in a population of 100 individuals that regularly mix with the wider community over the period of a month.

**Appendix C. The Airborne Transmission Model and the Estimation of $R_l$ and $R_n$**

For the classroom, Section 2.1.1, the model Equations (1) and (2) have been used to determine the concentration of viral particles and the infection risk in a classroom, respectively. These values were then used to determine the local/non-local concentration and infection risk, Tables A5 and A6, respectively. (We consider three ventilation settings, as described in the main text.) Here, we give more details about the way those values were generated. For the ADR Equation (1), we assume, as in [44], that

$$S_{\text{inf}} = S_0\delta(x - x_0)\delta(y - y_0), \tag{A1a}$$

$$S_{\text{vent}} = -\lambda A, \tag{A1b}$$

$$S_{\text{deact}} = -\beta A, \tag{A1c}$$

$$S_{\text{set}} = -\sigma A, \tag{A1d}$$

where $S_0$ is the emission rate of infectious particles from the infectious person, $(x_0, y_0)$ are the coordinates of the infectious person, $\lambda$ is the air exchange rate (ACH) of the room, $\beta$ is the virus deactivation rate, and $\sigma$ is the gravitational settling rate. Assuming that the air is recirculating in the room (with magnitude $v$), as in Figure A3, we can reduce the problem in two spatial dimensions. Hence, Equation (1) becomes

$$\frac{\partial A}{\partial t} + v\frac{\partial A}{\partial x} - K\left(\frac{\partial^2 A}{\partial x^2} + \frac{\partial^2 A}{\partial y^2}\right) = S_0\delta(x - x_0)\delta(y - y_0) - (\lambda + \beta + \sigma)A. \tag{A2}$$

Equation (A2) has a semi-analytical solution which allows for swift simulations of the viral concentration in a classroom as a function of space and time. We use parameter values as in Table A4—see [44]. As in [44], we assume one infectious person at the centre of the room who may be just breathing or talking with or without a mask. For more details on the model, please see [44]. The associated code is available to download from the repository https://github.com/zechlau14/Modelling-Airborne-Transmission, accessed on 24 November 2021.

Here, in order to determine the local/non-local infection ratio ($c_l/c_n$), the viral concentration and the infection risk are determined at two locations: 2 m directly downstream from the infected person ('local') and 4 m vertically upwards from the infected person ('non-local'). We simulate for a period of 5 h, assuming this to be a reasonable length of time for studying transmission in a classroom. The simulations are conducted assuming a mask of 50% efficiency and three ventilation settings ranging from 'very poor' to the setting recommended by ASHRAE. We also consider three modes of activity for the infected person: 'breathing', 'low activity' which we take to be 20% talking and 80% breathing and, 'talking' (the latter might be more relevant for an infected educator). Table A5 contains the concentration values and Table A6 contains the infection risk for the aforementioned activity modes and the three ventilation settings, with or without a mask.

We then assume that the background $R$ number in [58] corresponds to 'Low activity' *and* 'very poor' ventilation. We then calculate the effective $R$ number according to the ratio of the local infection risk after 5 h with the NPI applied versus the local infection risk after 5 h in 'very poor' ventilation. We also calculate the local:non-local infection ratio ($c_l/c_n$) for each NPI from the local and non-local infection risk during low activity. We summarise these in Table A7.

Finally, in Table A8 we split the $R$ number into local and non-local numbers, $R_l$ and $R_n$ using the relationships $R_l = P_{Al}R/(P_{Al} + P_{An})$ and $R_n = P_{An}R/(P_{Al} + P_{An})$. Note that in relation to the main text terminology, $c_l = P_{Al}$ and $c_n = P_{An}$.

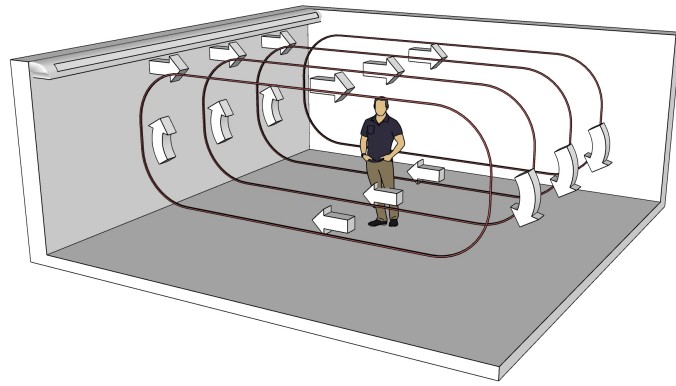

**Figure A3.** Recirculating airflow in the simulated classroom.

**Table A4.** Parameter values used for the estimation of $R_l$ and $R_n$ from the ADR Equation (A2) as developed in [44].

| Parameter | Value | Source |
|---|---|---|
| Classroom dimensions | $8\,\text{m} \times 8\,\text{m} \times 3\,\text{m}$ | |
| Airflow speed | $0.15\,\text{m/s}$ | [74] |
| Room air exchange rate (ACH) | Very poor ventilation: $0.12\,\text{h}^{-1}$<br>Poor ventilation: $0.72\,\text{h}^{-1}$<br>ASHRAE recommended ventilation: $3.00\,\text{h}^{-1}$ | [56]<br>[56]<br>[55] |
| Eddy diffusion coefficient | Very poor ventilation: $8.8 \times 10^{-4}\,\text{m}^2/\text{s}$<br>Poor ventilation: $5.3 \times 10^{-3}\,\text{m}^2/\text{s}$<br>ASHRAE recommended ventilation (good ventilation):<br>$2.2 \times 10^{-2}\,\text{m}^2/\text{s}$ | [59] |
| Breathing rate | $1.3 \times 10^{-4}\,\text{m}^3/\text{s}$ | [75] |
| Generation rate of infectious particles | Breathing: 0.5 particles/s<br>Low activity (20% talking, 80% breathing): 1.4 particles/s<br>Talking: 5 particles/s | [76] |
| Efficiency of mask | 0.5 | [57] |
| Virus deactivation rate | $1.7 \times 10^{-4}\,\text{s}^{-1}$ | [77] |
| Aerosol settling rate | $1.1 \times 10^{-4}\,\text{s}^{-1}$ | [78] |
| Median infectious dose | 100 particles | [79] |

**Table A5.** Steady-state local and non-local concentration of airborne contagions in secondary school classrooms with ventilation varying from 'very poor' ventilation to the ventilation recommended by ASHRAE. Calculated from the model in [44] with the parameters as in Table A4.

| | ACH = $0.12\,\text{h}^{-1}$ | | ACH = $0.72\,\text{h}^{-1}$ | | ACH = $3.00\,\text{h}^{-1}$ | |
|---|---|---|---|---|---|---|
| | $A_l$ | $A_n$ | $A_l$ | $A_n$ | $A_l$ | $A_n$ |
| **Breathing** | | | | | | |
| With mask | 21.1 | 3.7 | 8.2 | 4.3 | 3.1 | 2.0 |
| No mask | 42.2 | 7.3 | 16.4 | 8.6 | 6.2 | 4.1 |
| **Low activity (20% Talking and 80% Breathing)** | | | | | | |
| With mask | 59.4 | 10.2 | 23.0 | 12.0 | 8.7 | 5.8 |
| No mask | 118.3 | 20.5 | 46.0 | 24.1 | 17.5 | 11.5 |
| **Talking / Superspreader** | | | | | | |
| With mask | 202.8 | 35.1 | 78.8 | 41.3 | 30 | 19.8 |
| No mask | 422.4 | 73.1 | 164.2 | 86.1 | 62.5 | 41.2 |

**Table A6.** Local and non-local infection risk from airborne transmission after 5 h in a secondary school classroom with ventilation varying from 'very poor' ventilation to the ventilation recommended by ASHRAE. Calculated from the model in [44] with the parameters as in Table A4.

| | ACH = 0.12 h$^{-1}$ | | ACH = 0.72 h$^{-1}$ | | ACH = 3.00 h$^{-1}$ | |
|---|---|---|---|---|---|---|
| | $P_{Al}$ | $P_{An}$ | $P_{Al}$ | $P_{An}$ | $P_{Al}$ | $P_{An}$ |
| **Breathing** | | | | | | |
| With mask | 0.267 | 0.041 | 0.115 | 0.058 | 0.047 | 0.031 |
| No mask | 0.462 | 0.080 | 0.217 | 0.113 | 0.092 | 0.061 |
| **Low activity (20% Talking and 80% Breathing)** | | | | | | |
| With mask | 0.581 | 0.110 | 0.289 | 0.154 | 0.127 | 0.084 |
| No mask | 0.824 | 0.207 | 0.495 | 0.284 | 0.238 | 0.161 |
| **Talking / Superspreader** | | | | | | |
| With mask | 0.949 | 0.329 | 0.690 | 0.436 | 0.372 | 0.260 |
| No mask | 0.998 | 0.534 | 0.913 | 0.697 | 0.621 | 0.466 |

**Table A7.** The effective $R$ number and the local to non-local infection ratio, $c_l$:$c_n$ calculated from the low activity infection risk in Table A6 and assuming that the background $R$ number from [58] applies to the ACH = 0.12 h$^{-1}$ case.

| | ACH = 0.12 h$^{-1}$ | | ACH = 0.72 h$^{-1}$ | | ACH = 3.00 h$^{-1}$ | |
|---|---|---|---|---|---|---|
| | +No Mask | +Mask | +No Mask | +Mask | +No Mask | +Mask |
| Effective $R$ number | $R$ | $R/2$ | $3R/5$ | $3R/10$ | $2R/7$ | $R/7$ |
| Local to non-local infection ratio ($c_l$:$c_n$) | 3.98:1 | 5.20:1 | 1.74:1 | 1.88:1 | 1.47:1 | 1.51:1 |

**Table A8.** Local $R$ number, $R_l$, and non-local $R$ number, $R_n$, for airborne transmission from an infectious person with or without a mask in a secondary school classroom with ventilation varying from 'very poor' ventilation to the ventilation recommended by ASHRAE. Calculated from the relation $R_l = P_{Al}R/(P_{Al} + P_{An})$ and $R_n = P_{An}R/(P_{Al} + P_{An})$, the low activity infection risk presented in Table A6, the $R$ values presented in Table 2, and the scaling factor presented in Table A7.

| | ACH = 0.12 h$^{-1}$ | | ACH = 0.72 h$^{-1}$ | | ACH = 3.00 h$^{-1}$ | |
|---|---|---|---|---|---|---|
| | $R_l$ | $R_n$ | $R_l$ | $R_n$ | $R_l$ | $R_n$ |
| Without mask | 0.64–1.36 | 0.16–0.34 | 0.31–0.65 | 0.17–0.37 | 0.14–0.29 | 0.09–0.20 |
| With mask | 0.34–0.71 | 0.06–0.14 | 0.16–0.33 | 0.08–0.18 | 0.07–0.15 | 0.05–0.10 |

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
