# Peer review of "A General Computational Framework for COVID-19 Modelling with Applications to Testing Varied Interventions in Education Environments"

_covid, doi:10.3390/covid1040055_

Round 1

Reviewer 1 Report

The authors present their work regarding a construction of a spatially-compartmental, individual-based model on the spread of SARS-CoV-2 in indoor spaces with a particular emphasys on the role of lateral flow tests used as a mass screening tool within an educational setting. The topics is original and exceptionally important for the public health efforts directed towards attempts to control the pandemices. The study design, methods and results are clearly presented and the results are supported by data.

I recommend the manuscript for publication but it does require several minor revisions: 

  1. From the biological standpoint, it is exceptionally important to make a distinction between the virus and the disease in scientific publications. For example, in the first sentence of the Abstract "the spread of COVID-19" needs to be replaced with "the spread of SARS-CoV-2". Please make sure you consistently correct this issue (virus vs disease) in the manuscript, particularly because it is focused on the spread of infection.
  2. In section "Introduction", please delete the paragraph "Our results have been presented to.......policy planinning for the pandemics". I realy do not feel it adds to the manuscript in terms of scientific content. In case you feel the need to make this statement, if could be included into discussion to help the readers to understand the impact of your work. 
  3. In section Interventions, 2.1.1. scenario classroom, it would be useful for the readers to define the term "ventilation" more accurately (what are the criteria for good or bad ventilation, relation between the size of the classroom and the size of the windows, the issue of airconditioning, just provide us with a bit more information)
  4. Considering the results of your study regarding the false-positive results of lateral-flow assays, please provide a range of false-positive results for LFD described in the literature (in addition to ref 22) in order to illustrate the difference in the rate of false-positive results in LDTs from different manufacturers used worldwide. In addition, it would be very helpful to report the false-positive rate in specific LDTs used in your country obtained by independent laboratory-based validations. This might help the readers to put your results in an appropriate context and fully appreciate the importance of your work. 

Reviewer 2 Report

Moore et al., constructed a spatially-compartmental, individual-based model of the spread of Covid-19 in indoor spaces and provided an open-access and user-friendly online applet that simulates the individual-based model, complete with user tutorials to encourage the use of the model to aid educational policy decisions as input infection data evolves (https://bit.ly/CV19 INTER IBM).  Basic questions need to be addressed.

  • In this study they considered “student populations mixing in a classroom and in halls of residence” , What do they mean here?
  • In the introduction, it is mentioned as (e.g. classrooms or halls of residence). , what about the students who come from home, spreading level is possible high when the students mix with other communities?
  • “Hence, there is still an urgent need for continuing to study and compare NPIs to identify their optimal use, with emphasis on educational environments. “ how this model will help to plan future prevention
